🔓 | **Open Peer Review** | Antimicrobial Chemotherapy | Research Article

# Dietary impact on the gut resistome: western diet independently increases the prevalence of antibiotic resistance genes within the gut microbiota

Robert Keskey,[1] Stijn Bluiminck,[2] Naseer Sangwan,[3] Rebecca Meltzer,[1] Adam Lam,[1] Renee Thewissen,[2] Alexander Zaborin,[1] Harry van Goor,[2] Olga Zaborina,[1] John Alverdy[1]

**ABSTRACT** Approximately half of surgical site infections are caused by pathogens resistant to the antibiotics used for prophylaxis. We recently demonstrated that when mice are fed a western diet (WD) that is high in fat and low in fiber, exposed to antibiotics, and undergo an otherwise recoverable surgery, they develop lethal sepsis associated with dissemination of multi-drug-resistant pathogens. Here, we hypothesized that a WD alone can drive the intestinal microbiome to become populated by antibiotic-resistant bacteria independent of exposure to antibiotics. The cecal microbiota response to antibiotics was determined utilizing Biolog Phenotype Microarrays in the presence of 48 different antibiotics. WD-fed mice had a significant increase in antibiotic resistance within their microbiome compared to mice on a standard low-fat, high-fiber diet (SD) including aminoglycosides, tetracyclines, cephalosporins, fluoroquinolones, and sulfamethoxazole. By metagenomic sequencing, there was an increase in the antibiotic resistance genes (ARGs) within the WD cecal microbiota, including *CfxA2*, *ermG*, *TetQ*, and *LnuC*. After just 7 days of WD, the ARGs *ermG* and *CfxA2* were detectable within the stool. WD independent of antibiotic exposure increases the presence of ARGs within the gut microbiota independent of antibiotic exposure.

**IMPORTANCE** Antibiotic resistance is a major challenge in healthcare and results in significant morbidity and mortality. Currently, half of surgical site infections are caused by pathogens resistant to antibiotics used for prophylaxis. In this study, we demonstrate that a western diet alone has the ability to increase the presence of antibiotic resistance genes within the gut microbiome. By understanding dietary influences on the gut resistome, we may improve our understanding of infections with antibiotic-resistant organisms and one day develop personalized antibiotic regimens based on an individual's gut resistome.

**KEYWORDS** gut resistome, antibiotic resistance, western diet

**Peer Reviewer** Xinming Xu, Fudan University, Shanghai, China

Address correspondence to John Alverdy, jalverdy@bsd.uchicago.edu, or Robert Keskey, rkeskey@uchicagomedicine.org.

Robert Keskey and Stijn Bluiminck contributed equally to this article. Robert Keskey is listed first as he handled the final submission and edits of the manuscript.

Harry van Goor, Olga Zaborina, and John Alverdy contributed equally to this article.

The authors declare no conflict of interest.

See the funding table on p. 10.

The rise of antibioti- resistant organisms continues to plague the healthcare system resulting in over 3 million infections and almost 50,000 deaths per year (1, 2). Antibiotics have played an essential role in the advancement of medicine and improvement in treatment and prevention of infectious diseases; however, the efficacy of antibiotics in the era of antibioti-resistant organisms remains tenuous, with an increasing number of antibioti-resistant organisms responsible for infections, resulting in increased hospitalizations, morbidity, and death.

The intestinal microbiota is an important reservoir for pathogens and antibiotic resistance genes (ARGs) (3, 4). The role of environmental exposures in increasing the presence of ARGs within the microbiota has been repeatedly demonstrated across

multiple studies (5, 6). In human studies, it appears that ARGs in gut microbiota are dependent on prior antibiotic exposures, diet, environment, and physiological stressors including surgical intervention (7–9). These studies allude to the influence of a western diet (WD), high in fat and low in fiber, in the development and accumulation of ARGs within the microbiota. Human studies have also demonstrated associations between diet and the gut resistome (7, 10). A recent twin study out of the UK demonstrated that host genetics only accounted for 25% of the variation in the gut resistome, further supporting the dominant effect of environmental exposures on the gut resistome (11). However, the mechanism and ability of diet alone to increase the presence of ARGs within the microbiota remains unexplored.

The gut resistome is directly influenced by environmental exposures, which can induce changes to the resistome that are long-lasting. A human study looking at the gut resistome of swine farm workers demonstrated that the workers developed a gut resistome directly influenced by their exposure to the swine farm and persisted after the workers moved to a new environment (5). Additionally, pre-term infants who are admitted to the NICU and require antibiotic treatment have been shown to have persistent colonization with antibioti-resistant organisms despite recovery and maturation of their microbiota after discharge (12). To further implicate the role of diet and environment in the gut resistome, geographic and diet influences have been shown to shape the resistome of chimpanzees and humans alike. Captive apes tend to have a higher relative abundance of antibioti-resistant genes within their gut resistome when compared to the resistome of wild apes (13). Similarly, humans who have been westernized and engage in agriculture have a significant increase in ARGs compared to hunter and gatherer tribes (14). It is clear that environmental exposures, including western diet, shape the intestinal microbiota and lead to the accumulation and persistence of ARGs.

A WD, high in fat and low in fiber, has long been shown to result in significant perturbations to the intestinal microbiota, changing their functionality (8, 15–18). The disturbance of the intestinal microbiota by a WD has been demonstrated to have a significant impact on how the intestinal microbiota respond to host stressors, including antibiotic exposure, infection, and surgery (8, 19). Here, we study the effect of WD on the development of antibiotic resistance within the gut microbiota. We demonstrate that WD alone results in an increased antibiotic resistance of gut microbiota to several classes of antibiotics.

## MATERIALS AND METHODS

### Mouse experiments

Male 6-week-old C56BL/6 (Charles River Laboratory) mice were housed within a temperature-controlled, 12-h light/dark cycled room of the animal facility of the University of Chicago. Mice were housed in a barrier facility constructed to ensure the prevention of adventitious infectious agents.

To avoid stress from social isolation, mice were housed at five mice per cage. The weights of all mice were monitored weekly. All experiments were performed in accordance with the National Institutes of Health guidelines, and approval was obtained from the University of Chicago Animal Care and Use Committee (Protocol 71744). Six cages of C56BL/6 mice were randomly assigned to two experimental groups: (i) Standard chow diet-fed mice ($n = 14$) and (ii) WD-fed mice ($n = 25$). The mice were assigned to *ad libitum* feeding with either a WD (60% kcal fat, 0 g fiber; Bio Serve, mouse high-fat diet, cat#S3282) or standard diet (SD, 18% kcal fat, 18 g fiber; standard mouse chow; Envigo) for 7.5 weeks. Afterward, all mice were sacrificed by $CO_2$ inhalation for 5 min. After the sacrifice, a laparotomy was performed to remove the cecum within an anaerobic chamber. Approximately 25% of the freshly collected cecal content was analyzed by Biolog Phenotype MicroArray. A small portion of the freshly collected cecal content was put on a pH strip (Sigma-Aldrich; Hydrion Brilliant pH dipsticks #Z264784). The remaining

cecal contents were collected in 10% glycerol at the time of sacrifice and stored at –80°C until use for liquid culture experiments. For additional cecal microbiota and stool microbiota comparison, fresh stool samples were collected right before the sacrifice of 5 WD- and 5 SD-fed mice.

## Biolog

The Phenotype MicroArray panels 11C and 12B (Biolog, Hayward, 398 CA, USA) were used to determine the antibiotic-resistance potential by measuring the metabolic activity as a degree of redox reduction due to respiration in the presence of four concentrations (low to high) of 48 different antibiotics. Freshly collected cecal content or stool samples were homogenized in normal saline and afterward filtered using 70 µm DB Falcon Cell Strainers. An aliquot of the filtered sample was introduced to 16 mL IF-0a GN/GP base inoculating fluid (Biolog). The optical density (OD) was adjusted to an OD of 0.025 using the OD 600. Afterward, 7.5 mL of IF-0a bacterial solution was introduced to 100 mL of IF-10b GN/GP base inoculating fluid (Biolog), 1.2 mL Dye mix D (anaerobic bacteria) or 1.2 mL of Dye mix H (aerobic bacteria) (Biolog), 600 µL 1 M glucose solution and 10.7 mL sterile demineralized water. Each sample was plated on 96-well Phenotype Microarray Plate 11C and 12B that cover a range of common and clinically relevant antibiotics, as well as other antimicrobial compounds (Biolog Hayward, CA). Anaerobic plates were sealed with a PCR film within the anaerobic chamber. Afterward, both aerobic and anaerobic plates were incubated for 24 h at 33°C in the OmniLog incubator/reader (Biolog). Mann-Whitney *U*-tests were conducted for comparison between WD and SD samples utilizing Bonferroni correction for multiple hypothesis testing.

## qPCR

DNA was isolated from approximately 100 mg of stool and cecal samples using the DNeasy kit. Approximately 25 ng of DNA was utilized for quantitative PCR using Biorad SYBR Green following the manufacturer's protocols. Primers were utilized for ARGs of interest *CfxA2* (Fwd, GCAAGTGCAGTTTAAGATT; Rev, GCTTTAGTTTGCATTTTCATC) and *ErmG* (Fwd, GTGAGGTAACTCGTAATAAGCTG; Rev, CCTCTGCCATTAACAGCAATG). Values were normalized by subtracting the 16S rRNA cycle threshold (Ct) value from the Ct values for the target gene to calculate ΔCt values, which are expressed as $2^{[Ct\,(16S\,PCR)\,-\,Ct\,(target\,PCR)]}$.

## Shotgun metagenomic sequencing and bioinformatics analysis

Picogreen (Invitrogen) was used to quantify DNA concentration in samples. DNA was sheared using a Covaris, and libraries were constructed with the Nugen Ovation Ultralow Library protocol. We aimed for an insert size of 400 bp to maximize data. Amplified libraries were visualized on an Agilent Bioanalyzer DNA1000 chip, pooled at equimolar concentrations based on these results, and size selected using a Sage Blue Pippin 1.5% cassette. The library pool was quantified using a Kapa Biosystems qPCR protocol, then sequenced on the Illumina NextSeq in a 1 × 150 paired-end sequencing run using dedicated read indexing. The samples were demultiplexed with bcl2fastq (raw sequences, SRA PRJNA1148150). Post-sequencing quality control, trimming, and filtering were performed using Nesoni (https://github.com/Victorian-Bioinformatics-Consortium/nesoni). Next, we aligned the high-quality reads to the mouse genome using Gencode version GRCm38 using STAR aligner. Reads mapped to the human genome were excluded for further analysis. Non-mouse reads were processed for taxonomical and functional analysis using MetaPhlan3 (20) and HUMAnN3 (21), respectively. Quality-filtered reads were assembled into contigs using MetaBAT2 (22). Antimicrobial resistance genes (ARGs) were predicted from assembled metagenomic contigs using Abricate v12 (23, 24). Abricate was run against the CARD database (25). The parameters were set at --sensitive, --min-id 80, --min-cov 90. Results were filtered to include only hits with a minimum alignment length of 100 bp. This approach provided a comprehensive

and robust assessment of ARG diversity within the metagenome. The output was parsed to generate a summary table containing information on the identified ARGs, including contig ID, gene name, database source, percentage identity, percentage coverage, and alignment length. The median sequencing depth and IQR (73,742,246 ± 10,836,546) A pairwise comparison was conducted utilizing White's non-parametric *t*-test (26).

## Culture analysis

The cecal content of five WD-fed and five SD-fed mice was collected and stored in 10% glycerol at −80°C until use. The sample was homogenized in normal saline and afterward filtered using 70 µm DB Falcon Cell Strainers. The OD of the sample was adjusted to an OD of 0.01 in 5 mL tryptic soy broth (TSB) and 5 mL of Clostridial Broth. 180 µL of the sample was put on a 96-well plate. 20 µL of the antibiotics was added in a threefold increasing concentration rate: amikacin (10 mg/L), ofloxacin (1/3/5 mg/L), neomycin (1/5/10 mg/L), and cefoxitin (1/5/10 mg/L). Prior pilot results determined the concentration of 10 mg/L for amikacin. The anaerobic plates were tightly sealed with PCR film and kept inside the anaerobic chamber at 37°C. Aerobic plates were kept outside the anaerobic chamber and placed in a 37°C incubator. The OD was measured at 0, 16, and 24 h after inoculation. After 24 h, the control liquid culture samples were placed on Columbia Nalidixic Acid Agar (CNA agar) and MacConkey plates containing antibiotics at a concentration determined from the liquid culture results (amikacin: 1 mg/L, neomycin: 10 mg/L, cefoxitin: 10 mg/L, and ofloxacin: 1 mg/L). Growing colonies were isolated and used for species and susceptibility identification.

## RESULTS

### WD results in weight gain, acidification of gut microenvironment, and significantly alters gut microbiota response to antibiotics

As expected, mice feeding on a WD significantly increased in overall weight (29.8% ± 3.1% vs 72.1% ± 11.3%, $P = 2.1e-5$, SD $n = 9$, WD $n = 15$, Fig. S1A) and resulted in alterations in the intestinal microenvironment demonstrated by a reduction in cecal pH (6.88 ± 0.4 vs 6.33 ± 0.23, $P = 2.3e-4$, SD $n = 9$, WD $n = 15$, Fig. S1B). Biolog phenotype antimicrobial microarrays were used to characterize how diet impacted the cecal microbiota resistome. The cecal microbiota resistance profile was inferred from the level of metabolic activity in the presence of the highest concentrations of antibiotics on the Biolog microarrays. Additionally, the basal metabolic activity was determined at the lowest concentrations of antibiotics, and the antibiotic-induced metabolic activity was normalized to basal metabolic activity. When the cecal microbiota was compared between standard chow diet (SD)- and WD-fed mice under anaerobic (Fig. 1A) and aerobic (Fig. 1B) conditions, there were 23 individual antibiotics that had significantly increased metabolic activity within WD microbiota under anaerobic conditions (Fig. 1A), and 25 individual antibiotics under aerobic conditions (Fig. 1B). The increased metabolic activity in the presence of antibiotics was similarly found to be present in the stool microbiota of WD-fed mice (Fig. S2). The antibiotic resistance profile based on Biolog was similar between the cecal and stool microbiota with a significant increase in metabolic activity in the presence of 13 different antibiotics. The PCA analysis of total antibiotic-induced metabolic activity revealed the distinct clustering between groups of mice feeding on SD and WD in both stool and cecal microbiota (Fig. 1C). To validate the antibiotic resistance noted on Biolog, cecal microbiota from WD and SD mice were grown in liquid culture in the presence of individual antibiotics. The WD cecal microbiota demonstrated a significant increase in growth in the presence of amikacin, cefoxitin, and ofloxacin. Additionally, WD cecal microbiota had an increase in growth in the presence of neomycin that did not reach significance (Fig. 1D).

The WD cecal microbiota Biolog activity was compared between different classes of antibiotics (Table S1). The WD microbiota demonstrated significantly increased activity in the presence of a variety of classes of antibiotics under both aerobic (Fig. 2A) and

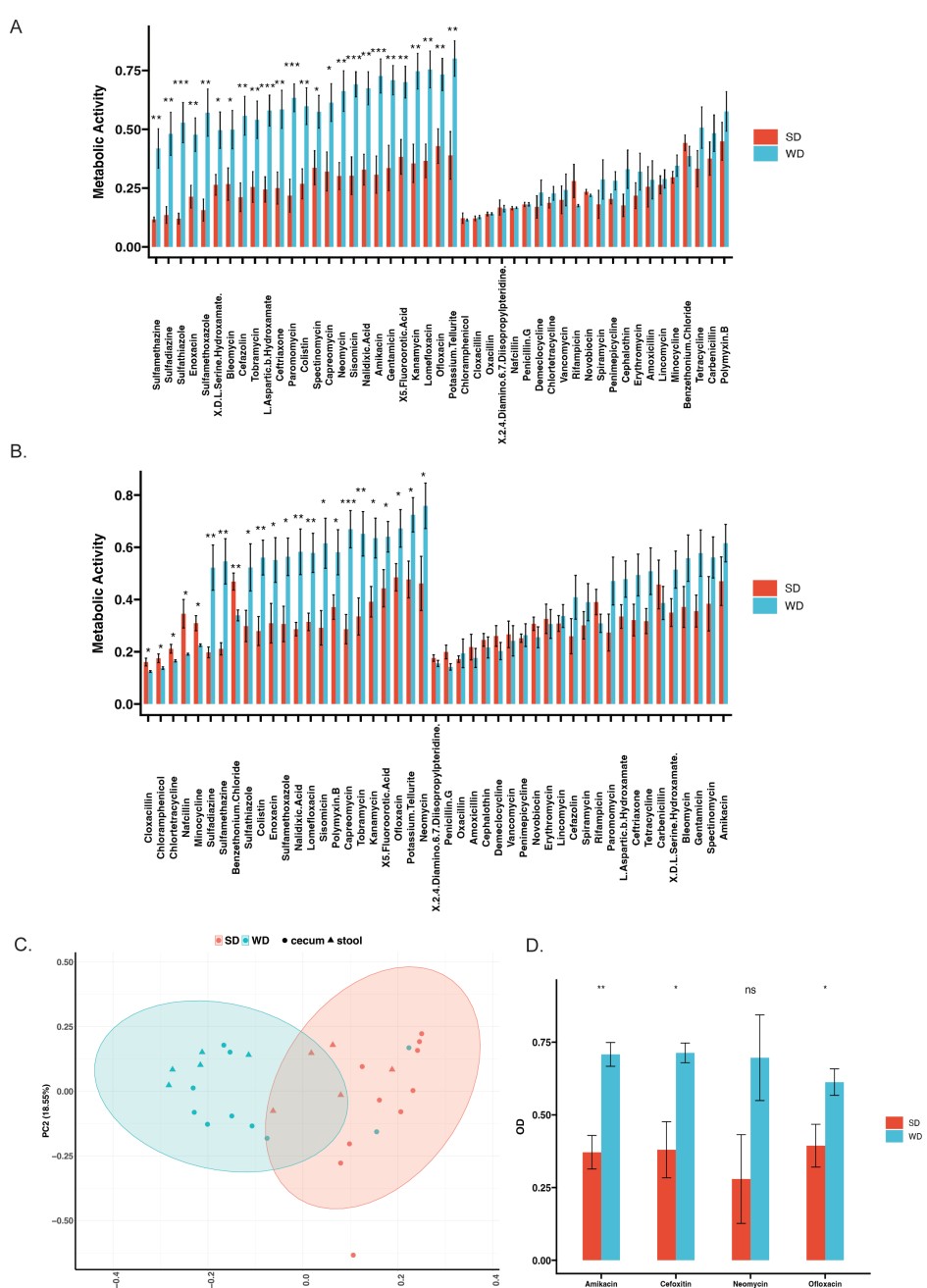

**FIG 1** WD results in a significant increase in the gut resistome. When the overall metabolic activity of the cecal microbiota was compared across all antibiotics on PM plates, there was a significant increase in metabolic activity across individual antibiotics in aerobic (A) and anaerobic (B) conditions ($n = 10$ WD, $n = 9$ SD). When comparing WD resistance profile across diets and between cecum and stool, there was clustering of samples by diet on principal component analysis (C). When cecal contents were cultured in the presence of antibiotics, there was a significant increase in growth among the microbiota isolated from western diet-fed mice. The resistance of the cecal microbiota to individual antibiotics on PM plates under aerobic (A) and anaerobic conditions (B). Cecal microbiota from WD ($n = 5$ with two technical replicates) and SD ($n = 5$ with two technical replicates) were grown in TSB liquid culture for 16 h in the presence of cefoxitin (10 mg/L), amikacin (10 mg/L), ofloxacin (1 mg/L), and neomycin (10 mg/L) (D). OD was normalized to the starting OD. Consistent with the Biolog results, there was a significant ($P < 0.05$) increase in growth of the WD microbiota in the presence of all antibiotics except for neomycin, which approached significance ($P = 0.08$). *$P < 0.05$, **$P < 0.01$, ***$P < 0.001$, ****$P < 0.0001$.

anaerobic (Fig. 2B) conditions, including aminoglycosides, cephalosporins, fluoroquinolones, and tetracyclines under aerobic conditions and sulfonamides, aminoglycosides, and fluoroquinolones under anaerobic conditions.

## Metagenomic sequencing reveals the accumulation of ARGs within the WD-fed cecal microbiota

Given the differences seen on Biolog between the cecal microbiota, shotgun metagenomic sequencing was performed to determine if the differences in metabolic activity seen on Biolog correlated with the antibiotic resistome of the cecal microbiota. On compositional analysis, WD feeding resulted in a reduction in both alpha (Fig. 3A) and altered beta diversity (Fig. 3B) in the stool and cecal microbiota when compared to SD-fed mice. We have also noticed the differences in composition between stool and fecal microbiota in WD-fed mice as compared to SD-fed mice (Fig. 3B). When the composition of the microbiota was compared between WD- and SD-fed mice, there was a significant increase in *Bacteroides vulgatus* within the cecum and stool of WD-fed mice (Fig. 3C and D), which is a normal gut commensal that has been shown to have pathogenic potential (27). SD-fed mice were found to have a large percentage of *Bacteroides ovatum,* which is a normal gut commensal that plays an important role in stimulating fecal IgA levels.

When determining the relative abundance of ARGs, four different ARGs were determined to be significantly increased in WD cecal microbiota compared to SD cecal microbiota: *ErmG*, *CfxA2*, *mel*, and *mefA* (Fig. 4A). Six ARGs were detected to be increased in the stool of WD-fed mice, with *CfxA2, mel*, and *tetQ* remaining undetectable in the SD stool and *ErmG*, *tetX*, and *tetO* at significantly lower levels (Fig. 4B). To determine whether these ARGs were present prior to the beginning of our experiments, time 0 stool and cecal microbiota were analyzed. For WD-fed mice, *lnuC*, *mel*, and *tetQ* were not present at time 0. *CfxA2* and *ErmG* were present at time 0 and increased in frequency with WD feeding. Similarly, for SD-fed mice, *CfxA2* and *ErmG* were detected at time 0 but decreased in abundance during the course of the experiment. To summarize, there are six ARGs that are significantly increased in the WD stool and cecal microbiota without prior exposure to antibiotics. There is a subset of ARGs that appear to be inherent to the mouse microbiota at the beginning of the experiment and increase with WD feeding. An additional subset was not detectable on sequencing at the beginning of the experiment but increased over time, indicating the ability of diet to alter the gut microbiota resistome. It is important to note that in the SD fed mice, the resistome appears to be isolated to the stool microbiota with complete absence of the WD-associated ARGs in the cecum (Fig. 4A and B). To confirm the presence and relative abundance of these ARGs, PCR was performed on stool collected from WD and SD mice from day 1 and day 7 after initiation of their respective diets. There was a significant increase in *tetQ*, *ErmG*, and *tetW* gene levels in the stool of WD-fed mice after 1 day of WD feeding (Fig. 4C). All four genes tested had a significant increase in expression in WD-fed mice after 1 week of WD feeding (Fig. 4C).

To further characterize the prevalence of ARGs within individual species within the microbiota, the ARGs were binned to metagenomes and the species harboring the ARGs were determined by BLAST. Across the mice, the WD-associated ARGs were detected in *Bacteroides dorei* (*ErmG*, *mel*), *Bacteroides fragilis* (*ErmG*), *Parabacteroides merdae* (*CfxA2*), *Mitsuokella sp. AF33-22* (*lnuC*), *Bacteroides acidifaciens* (*tetQ*), *Lactobacillus reuteri* (*lnuC*), and *Clostridium culturomicum* (*mefA*). The species harboring these genes are typical Firmicute and Bacteroides species of the microbiota.

## DISCUSSION

Here, we demonstrate that adherence to a WD independently results in an increase in the prevalence of ARGs and development of the gut resistome without exposure to antibiotics. Prior studies have demonstrated that WD is associated with the development of antibiotic resistance within the gut microbiota but have recognized the many

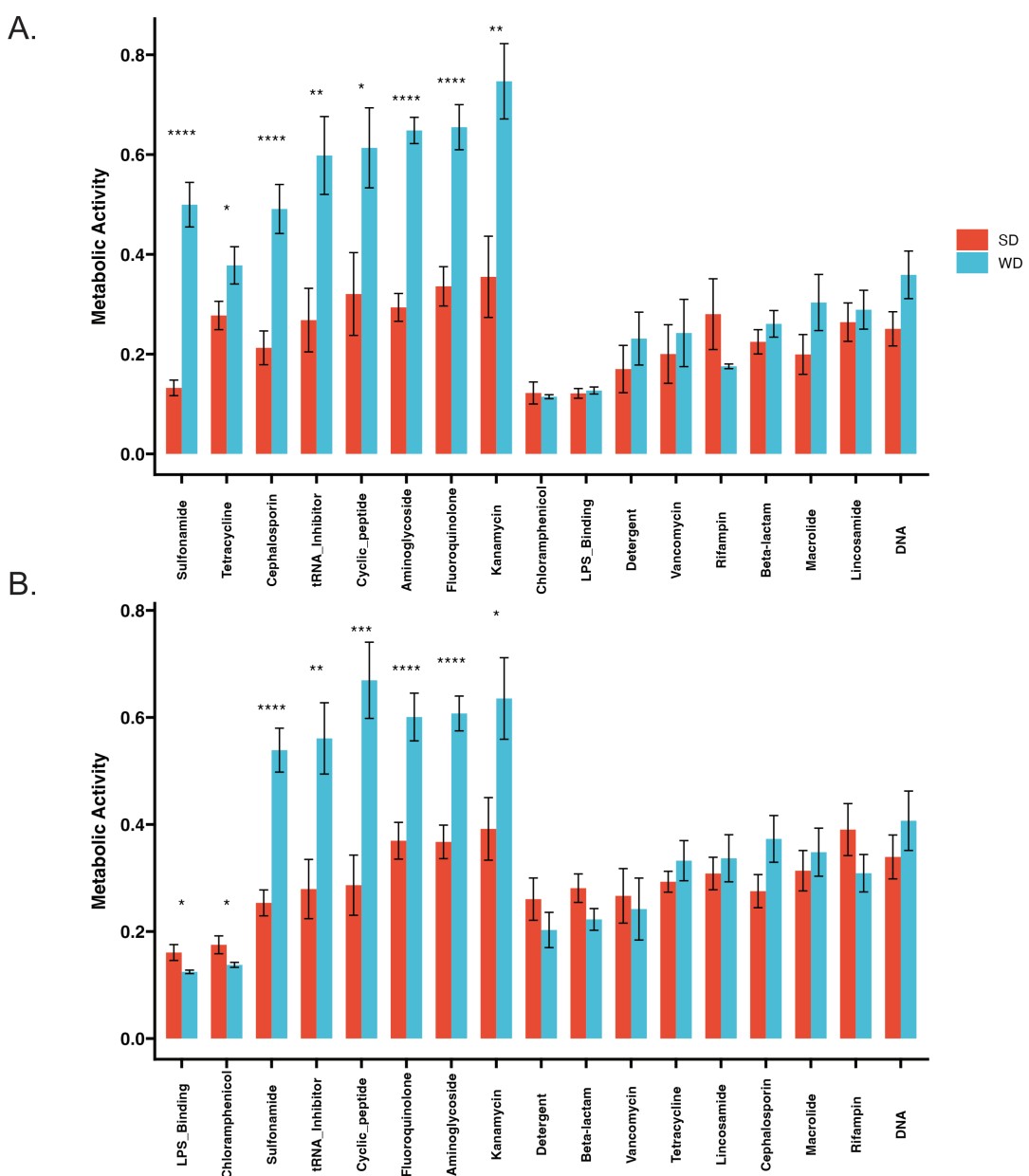

**FIG 2** WD microbiota maintain resistance across antibiotic classes. Biolog activity from WD and SD mice under aerobic (A) and anaerobic (B) conditions was compared, grouping antibiotics by antibiotic class. *$P < 0.05$, **$P < 0.01$, ***$P < 0.001$, ****$P < 0.0001$.

confounding environmental factors (7, 9, 10). This is the first study to demonstrate a diet-dependent alteration in the gut resistome when all other environmental and genetic variables are held constant. Furthermore, the gut resistome profile appeared to be sensitive to dietary interventions with significant alterations of the resistance profile within days of dietary changes. These findings are important as we begin to understand how diet and environmental exposure may explain the risk of colonization with antimicrobial-resistant pathogens and the response of the gut microbiota to antibiotics.

To our knowledge, this is the first study that utilized Biolog metabolics along with metagenomic sequencing to characterize the gut resistome. There was remarkable consistency between the findings seen on metagenomics and Biolog phenotype microarrays. For example, *CfxA2* (ARG responsible for cephalosporin resistance) was found on metagenomic to be increased in the cecal and stool microbiota of WD mice and paralleled similarly with an increased metabolic activity of WD cecal microbiota in

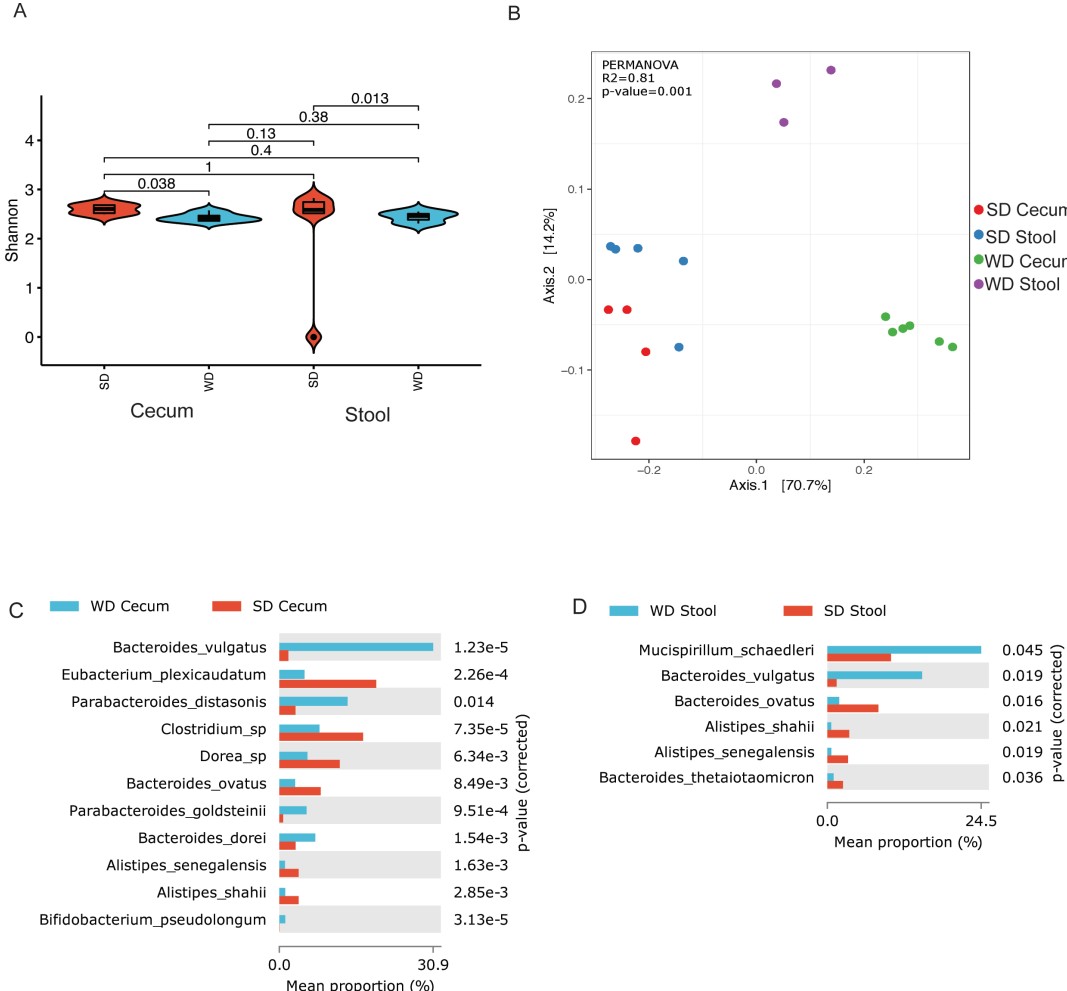

**FIG 3** WD affects the composition of cecal and stool microbiota. Analysis of metagenomics sequencing revealed significant differences in cecal and stool microbiota composition. WD cecum and stool had significantly lower alpha diversity (A) when compared to SD cecum and stool microbiota. Furthermore, there was distinct clustering by diet and location (stool vs cecum) by beta diversity by Bray-Curtis dissimilarity (B). The most significantly different abundant species between SD and WD cecal and stool microbiota are displayed in panels C and D, respectively.

the presence of cephalosporins. Similarly, the presence of *ErmG*, *mel*, *and mefA* is associated with macrolide resistance, and WD cecal microbiota was found to have a significant increase in metabolic activity in the presence of macrolides on the Biolog assays. The consistency between the metabolic and metagenomic assay strengthens the conclusions made about WD alterations in the gut resistome found in this study.

The accumulation of ARGs seen within *Parabacteroides* is consistent with prior research (28–30). Bacteria from the phylum *Bacteroidetes* are common gut commensals but can act as pathogens and are responsible for many anaerobic infections (29). More importantly, many infections treated in the hospital are the result of mixed flora, both aerobic and anaerobic, and require appropriate antibiotics targeting the anaerobic bacteria. Extensive studies throughout Europe have demonstrated that treatment failure often occurs due to antimicrobial resistance amongst anaerobic species (31). An observational study in humans demonstrated a significantly high level of antimicrobial resistance within anaerobic commensal species including resistance to clindamycin and metronidazole, antimicrobials utilized to empirically treat anaerobes (32). These findings, in conjunction with this study, emphasize the importance of understanding the dietary impact on the gut resistome and its future role in guiding antibiotic therapies.

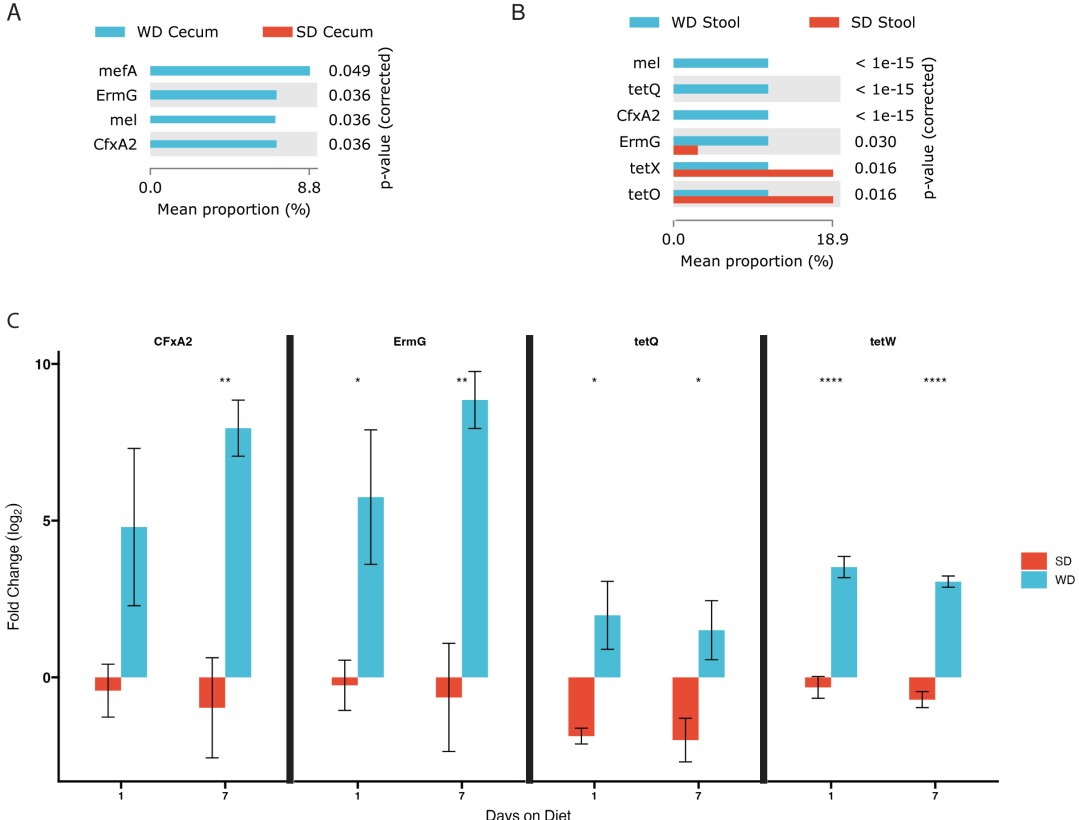

**FIG 4** Antibiotic resistance genes (ARGs) were identified within the metagenomes using the Comprehensive Antibiotic Resistance Database (CARD). There was a significant increase in the abundance of ARGs in both the cecal (A) and stool (B) microbiota of WD-fed mice. PCR was used to determine the presence and accumulation of ARGs *tetQ*, *CfxA2*, *ErmG*, and *tetW* (C) within the stool of WD- and SD-fed mice after 7 days of being on their respective diet (*n* = 5 per group). *$P <$ 0.05, **$P <$ 0.01, ***$P <$ 0.001, ****$P <$ 0.0001.

It remains unclear how the presence of these ARGs impacts both the host and gut microbiota in the setting of antibiotic exposure. Clinically, antibiotic resistance testing accounts for single species' response to antibiotics, but bacteria within the gut form complex communities of many different species that have co-evolved strategies to deal with environmental pressures. Previous studies have demonstrated that when communities of bacteria are exposed to antibiotics, there is a selection for genes that engage in cooperative drug resistance (33). It is unlikely that horizontal gene transfer occurs across phyla of bacteria; however, it is plausible that accumulation of ARGs even within "non-pathogenic" bacteria may have the ability to act as an antibiotic sink and shield potential pathogenic bacteria from antibiotic exposure (34). A recent study demonstrated that mice colonized with beta-lactamase-producing *Escherichia coli* and treated with beta-lactam antibiotics had higher colonization with beta-lactam-sensitive pathogens, *Listeria monocytogenes* and *Clostridium difficile*, suggesting that antibiotic-resistant pathogens within the gut can shield potential pathogens from antibiotics (34). Additionally, these antibiotic-resistant bacteria may have the ability to outcompete other native microbiota species commonly associated with the promotion of health. Given the large role diet plays in shaping both the composition of the intestinal microbiota and the antibiotic resistance profile, the dietary impact on the gut resistome should be considered when providing patients with empiric antibiotic treatment against gut-derived pathogens. Currently, when antibiotic prophylaxis is used or treatment of a gut-derived pathogen is carried out, there is little attempt to individualize the antibiotic regimen for the individual's gut microbiota. Our data suggest that empiric antibiotics against gut pathogens may benefit from being personalized to the resistance profile of an individual's gut resistome,

and dietary interventions may provide a means of changing the gut resistome to alter pathogen susceptibility and improve the success of antibiotic treatment.

Our study is not without limitations; it was conducted in a single strain of mice from a single vendor. It is clear that both genetics and environment contribute to the gut resistome, and it is likely that both of these factors may result in different results than demonstrated here. Finally, biolog and liquid cultures are artificial growth environments that in turn select for organisms capable of growing in these culture conditions and may result in the loss of certain species of bacteria that may also contribute to the gut resistome *in vivo*. To account for these limitations, metagenomic sequencing provided a snapshot of the presence of ARGs in uncultivable bacteria.

## Conclusion

Here, we show WD independent of antibiotic exposure results in the increased presence of ARGs within the gut microbiota. As we continue to understand the contributing factors to the gut resistome, we may be able to advance care to curtail the rise of antibiotic resistance. Through dietary intervention and the use of antibiotic regimens personalized to an individual's gut microbiota, we may more efficaciously treat gut-derived infections as well as eliminate the use of empiric antibiotic selection that may be ineffective depending on an individual's gut resistome.

### ACKNOWLEDGMENTS

This work was accomplished with the help of Jason Koval, who assisted with running the Biolog assays, and Hilary Morison, whose lab assisted with completion of the metagenomic sequencing.

This work was supported by NIH grant R01GMO62344-22.

R.K.: Conceptualization, Methodology, Investigation, Formal analysis, Writing—original draft, Visualization, Project administration. S.B.: Conceptualization, Investigation, Formal analysis, Writing-original draft. N.S.: Visualization, Data analysis, Writing-original draft. R.M.: Conceptualization, Formal analysis, Methodology, Writing – original draft, Writing – review and editing. A.L.: Methodology, Investigation, Writing—review and editing. A.Z.: Methodology, Investigation. R.T.: Methodology, Investigation. H.V.G.: Conceptualization, Writing—review and editing. O.Z.: Conceptualization, Methodology, Formal analysis, Writing—original draft, Visualization, Project administration; J.A.: Conceptualization, Methodology, Writing—original draft.

### AUTHOR AFFILIATIONS

[1]Section of General Surgery, Department of Surgery, University of Chicago, Chicago, Illinois, USA
[2]Department of Surgery, Radboud University Medical Center, Nijmegen, Netherlands
[3]Cleveland Clinic, Cardiovascular and Metabolic Sciences, Cleveland, Ohio, USA

### AUTHOR ORCIDs

Robert Keskey (ID) http://orcid.org/0000-0003-4638-3572
John Alverdy (ID) http://orcid.org/0000-0003-1854-4551

### FUNDING

| Funder | Grant(s) | Author(s) |
| --- | --- | --- |
| HHS | NIH | OSC | Common Fund (NIH Common Fund) | R01GMO62344-22 | John Alverdy |

### AUTHOR CONTRIBUTIONS

Stijn Bluiminck, Conceptualization, Formal analysis, Writing – original draft, Writing – review and editing | Naseer Sangwan, Data curation, Formal analysis, Methodology,

Visualization, Writing – review and editing | Rebecca Meltzer, Conceptualization, Formal analysis, Methodology, Writing – original draft, Writing – review and editing | Adam Lam, Conceptualization, Data curation, Formal analysis, Methodology, Writing – review and editing | Renee Thewissen, Conceptualization, Data curation, investigation, Methodology, Writing – review and editing | Alexander Zaborin, Conceptualization, Data curation, Methodology, Writing – review and editing | Harry van Goor, Conceptualization, Supervision, Writing – original draft, Writing – review and editing | Olga Zaborina, Conceptualization, Formal analysis, Methodology, Supervision, Writing – original draft, Writing – review and editing | John Alverdy, Conceptualization, Formal analysis, Funding acquisition, Methodology, Resources, Supervision, Writing – original draft, Writing – review and editing.

## DATA AVAILABILITY

The raw biolog data assessing antimicrobial resistance were included as a supplemental figure. The metagenomic sequences are uploaded to SRA (PRJNA1148150).

## ETHICS APPROVAL

The animal studies used in this paper were approved by the University of Chicago IACUC.

## ADDITIONAL FILES

The following material is available online.

### Supplemental Material

**Fig. 1 (Spectrum02766-24-S0001.tif).** Western diet increases weight gain and alters cecal pH compared to standard diet.
**Fig. 2 (Spectrum02766-24-S0002.tif).** Western diet mice stool microbiota has increased metabolic activity in the presence of antibiotics compared to SD mice stool microbiota.
**Supplemental material (Spectrum02766-24-S0003.docx).** Supplemental figure and table legends.
**Table S1 (Spectrum02766-24-S0004.xlsx).** Abx classes.
**Table S2 (Spectrum02766-24-S0005.xlsx).** Metadata for metagenomics.

### Open Peer Review

**PEER REVIEW HISTORY (review-history.pdf).** An accounting of the reviewer comments and feedback.

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
