## [Reviewer comments · Microbiology Spectrum]

Microbiology Spectrum

Dietary impact on the gut resistome: western diet independently increases the prevalence of antibiotic resistance genes within the gut microbiota

Robert Keskey, Stijn Bluiminck, Naseer Sangwan, Rebecca Meltzer, Adam Lam, Renee Thewissen, Alexander Zaborin, Harry van Goor, Olga Zaborina, and John Alverdy

Corresponding Author(s): Robert Keskey, University of Chicago Division of the Biological Sciences

Review Timeline:

Submission Date:	October 31, 2024
Editorial Decision:	February 22, 2025
Revision Received:	April 2, 2025
Editorial Decision:	May 14, 2025
Revision Received:	May 21, 2025
Accepted:	July 2, 2025

Editor: Jinxin Liu

Reviewer(s): Disclosure of reviewer identity is with reference to reviewer comments included in decision letter(s). The following individuals involved in review of your submission have agreed to reveal their identity: Xinming Xu (Reviewer #1)

Transaction Report:

DOI: <https://doi.org/10.1128/spectrum.02766-24>

Re: Spectrum02766-24 (Dietary impact on the gut resistome: western diet independently increases the prevalence of antibiotic resistance genes within the gut microbiota)

Dear Dr. Robert Charles Keskey:

Thank you for the privilege of reviewing your work. Below you will find my comments, instructions from the Spectrum editorial office, and the reviewer comments.

Revision Guidelines

Sincerely,
Jinxin Liu
Editor
Microbiology Spectrum

Reviewer #1 (Comments for the Author):

Abstract

1. Resistance genes should be in italic. Please also check throughout the manuscript (especially in Results section and Figures).

Methods

-Is there any paper validated that high-fat diet could be defined as the western diet. Please cite some validation papers.

Otherwise, why not just say high-fat diet instead of western diet.

- There are two overlapping subsections: "shotgun metagenomic sequencing" and "shotgun metagenomic sequencing and bioinformatics analysis". Please restructure the content and remove the duplicated part.
- Method regarding "To further characterize how these ARGs were shaping the microbiota, the ARGs were binned to metagenomes and the species harboring the ARGs were determined by BLAST." was not described. Please add the related method in this section.
- Need to clarify how to define the "abundance" in the bioinformatics analysis section. It's really important. I think you should calculate the relative abundance of gut resistome to quantify.

Results

- Some figs and tables were not bolded in the text.
- When you say "As expected, mice feeding on a western diet significantly increased in overall weight...", please add the statistical test and corresponding P value in this sentence, also add the number of two groups. This also applies to the remaining part in the Results section.
- Page 10 line 269-274. This part is interesting to me. Could the authors develop this part into a new subsection within Results? Please also do some visualizations such as Sankey plot to quantitatively and clearly to trace these resistance genes to certain gut microbes.
- Personally I don't think ARGs could shape the microbiota, please rephrase the sentence.

Figures

- Please write in a standardized manner.

Reviewer #2 (Comments for the Author):

This is an interesting study investigating the effects of a Western diet on antibiotic resistance. More metabolic activity of fecal and cecal microbiota from mice fed a Western diet was observed in the presence of different antibiotics compared with microbiota from mice fed a standard diet. There was also a higher relative abundance of 6 antibiotic resistance genes (ARGs) in the metagenomes of mice fed a Western diet. In the future, it would be really interesting to test if the changes in antimicrobial resistance are reversible - i.e. does switching back to SD after WD decrease antibiotic resistance and how long does it take?

Methods

What statistical tests were used to determine significant differences in metabolic activity and ARG levels?

Line 127: What is the "reversal period"?

Line 140: mitochondrial?

Lines 175-181 are a repeat of 165-171.

Line 185: Should say "the mouse genome" and "Non-mouse reads," right?

How were differentially abundant ARGs identified?

Results

Suggestions to make the figures clearer/more impactful:

- Consider putting the antibiotics in figures 1A and 1B in the same order so that it is easier to compare between anaerobic and aerobic. Similarly, ordering the antibiotic classes in 2A and 2B the same and the ARGs in 4A and 4B the same, as much as the overlap allows, would facilitate comparison.
- I'd recommend keeping the order of aerobic and anaerobic results the same in Figures 1 and 2 (ie. Have figure 1A and 2A both be anaerobic results and 1B and 2B both be aerobic results).
- I'd also recommend a consistent (and color-blind safe) color scheme for the 4 sample types (WD stool, WD cecum, SD stool, SD cecum) throughout all figures.

Line 223: It looks like 13 (not 16) are significant in Figure S2.

Line 228: Figure 1D indicates that the higher growth from WD with neomycin is not significant.

Line 241: Rather than increased beta diversity with WD it is probably more accurate to say altered beta diversity.

Line 248: Figure 4A shows 4 (not 6) ARGs that differ by diet in the cecum.

Lines 249 and 526: This is relative abundance not abundance, right?

Line 251-252: According to Figure 4, the ARGs that were higher in stool were not all the same as those higher in cecum.

Line 258: Please clarify that there are 6 ARGs increased with WD in the stool and cecum combined.

Lines 266 and 268: I don't think gene "expression" was measured, but rather gene levels.

What do the error bars in figures 1, 2, 3A and 4C represent? Should there be error bars on figs 3C&D, 4A&B? How many mice are represented in the metagenomic figures (3 and 4A&B)?

What was the metagenomic sequencing depth? Were any samples excluded due to low sequence counts?

Were other ARGs (besides those in Figure 4) detected?

Figure 4A and 4B: Are these samples after 7.5 weeks on the specific diet (so in 13.5 week old mice)?

For Figure 4C, why were day 1 and day 7 selected if the animals were fed different diets for 7.5 weeks? I don't think it is required but it would be interesting to perform qPCR on day 0 and 7.5 week samples too. Why was tetW selected for qPCR (Figure 4C)

but not represented in Figure 4A or 4B?

The metagenomic sequence data is available from the NCBI's SRA (PRJNA1148150). However, the diet and timepoint for each sample is needed for reproducibility and usefulness (either in the SRA record or as a supplemental table in the manuscript).

Discussion

It would be interesting to include a discussion of how the metagenomic results fit with the metabolic data. For example, *mefA* and *mel* are both at higher relative abundances with WD which fits with greater (although not significantly greater) metabolic activity from WD with macrolide exposure because "Mel, a homolog of MsrA, is an ABC-F subfamily protein associated with macrolide resistance. It is expressed on the same operon as *mefA* and *mefE*, both MFS-type efflux proteins that confer macrolide resistance." (from <https://card.mcmaster.ca/ontology/36910>)

Line 263: I think it should say: "with complete absence of the WD-associated ARGs in the cecum"

Minor:

Line 272: Should "merdei" be "merdae"?

Line 293: Should "Parabacteroidetes" be "Parabacteroides"?

Line 314: antibiotics to antibiotic

Abstract

1. Resistance genes should be in italic. Please also check throughout the manuscript (especially in Results section and Figures).

Methods

-Is there any paper validated that high-fat diet could be defined as the western diet. Please cite some validation papers. Otherwise, why not just say high-fat diet instead of western diet.

- There are two overlapping subsections: “shotgun metagenomic sequencing” and “shotgun metagenomic sequencing and bioinformatics analysis”. Please restructure the content and remove the duplicated part.

- Method regarding “*To further characterize how these ARGs were shaping the microbiota, the ARGs were binned to metagenomes and the species harboring the ARGs were determined by BLAST.*” was not described. Please add the related method in this section.

-Need to clarify how to define the “abundance” in the bioinformatics analysis section. It’s really important. I think you should calculate the relative abundance of gut resistome to quantify.

Results

- Some figs and tables were not bolded in the text.

- When you say “*As expected, mice feeding on a western diet significantly increased in overall weight...*”, please add the statistical test and corresponding P value in this sentence, also add the number of two groups. This also applies to the remaining part in the Results section.

- Page 10 line 269-274. This part is interesting to me. Could the authors develop this part into a new subsection within Results? Please also do some visualizations such as Sankey plot to quantitatively and clearly to trace these resistance genes to certain gut microbes.

-Personally I don’t think ARGs could shape the microbiota, please rephrase the sentence.

Figures

-Please write in a standardized manner.

Reviewer 1: Resistance genes should be in italic. Please also check throughout the manuscript (especially in Results section and Figures).

Author Response: Thank you for the correction. We have updated the resistance genes and they are now italicized throughout the paper as suggested.

Reviewer 1: Is there any paper validated that high-fat diet could be defined as the western diet. Please cite some validation papers. Otherwise, why not just say high-fat diet instead of western diet.

Author Response: Thank you, this is an important clarifying question on how we described a high fat, low fiber diet. In the microbiota literature, a diet that is high in fat and low in fiber has been commonly referred to as a "western diet." Referring to the diet only by the high fat content does not accurately reflect the absence of microbiota accessible fiber seen in western diet due to the high degree of processing which has a very important impact on both the microbiota composition and function. To provide further support of utilizing western diet to describe a high-fat, low fiber diet, the following references have been added (**p. 4, line 107**):

17. Las Heras V, Melgar S, MacSharry J, Gahan CGM. The Influence of the Western Diet on Microbiota and Gastrointestinal Immunity. *Annu Rev Food Sci Technol.* 2022;13:489–512.

18. Turnbaugh PJ, Bäckhed F, Fulton L, Gordon JI. Diet-Induced Obesity Is Linked to Marked but Reversible Alterations in the Mouse Distal Gut Microbiome. *Cell Host & Microbe.* 2008;3:213–23.

Reviewer 1: There are two overlapping subsections: "shotgun metagenomic sequencing" and "shotgun metagenomic sequencing and bioinformatics analysis". Please restructure the content and remove the duplicated part.

Author Response: Thank you for catching this formatting error, the methods section on metagenomics has been consolidated into one single section to make the section clear (**p. 6-7, lines 164-193**)

Reviewer 1: Method regarding "To further characterize how these ARGs were shaping the microbiota, the ARGs were binned to metagenomes and the species harboring the ARGs were determined by BLAST." was not described. Please add the related method in this section.

-Need to clarify how to define the "abundance" in the bioinformatics analysis section. It's really important. I think you should calculate the relative abundance of gut resistome to quantify.

Author Response: The methods section was updated to describe how the metagenomes were binned into individual species AND the determination of the individual organisms containing the ARG were better described (p.7 lines 179-182). The ARG mean proportion in Figure 4 indicates the proportions of sequences within the metagenome that map to the respective ARG.

Reviewer 1: Some figs and tables were not bolded in the text.

Author Response: Thank you for pointing out the formatting issues, this has been updated as suggested.

Reviewer 1: When you say "As expected, mice feeding on a western diet significantly increased in overall weight...", please add the statistical test and corresponding P value in this sentence, also add the number of two groups. This also applies to the remaining part in the Results section.

Author Response: The p values and the number of mice per group have been added into the text as suggested. Additionally, the p values and the number of mice per group can also be found in the figure legends.

Reviewer 1: Page 10 line 269-274. This part is interesting to me. Could the authors develop this part into a new subsection within Results? Please also do some visualizations such as Sankey plot to quantitatively and clearly to trace these resistance genes to certain gut microbes.

-Personally I don't think ARGs could shape the microbiota, please rephrase the sentence.

Author Response: This is indeed very interesting findings within our study; however, a Sankey plot was not feasible to make in this section as there is some variation between individual metagenomic samples that makes a construction of a Sankey plot not feasible as suggested. We have reworded the sentence as suggested: "To further characterize the prevalence of ARGs within individual species within the microbiota"

Reviewer 1: Please write figures in a standardized manner.

Author Response: The figure references were updated in the text to consistently reference the figures in a standardized manner

Reviewer #2 (Comments for the Author):

This is an interesting study investigating the effects of a Western diet on antibiotic resistance. More metabolic activity of fecal and cecal microbiota from mice fed a Western diet was observed in the presence of different antibiotics compared with microbiota from mice fed a standard diet. There was also a higher relative abundance of 6 antibiotic resistance genes (ARGs) in the metagenomes of mice fed a Western diet. In the future, it would be really interesting to test if the changes in antimicrobial resistance are reversible - i.e. does switching back to SD after WD decrease antibiotic resistance and how long does it take?

Author Response: Thank you for the reviewer's inquiry as we agree that determining if the dietary alterations the gut resistome are reversible is a very important line of inquiry. In our previous papers, we have demonstrated that functional and compositional changes to the microbiota induced by WD feeding are reversible with SD feeding. Unfortunately, determining the exact duration of the dietary reversal how it impacts the gut resistome as well as the molecular details associated with dietary reversal require a significant amount of experimental inquiry that go beyond the scope of this current publication that are currently ongoing and not ready for publication, but we really appreciate the comment.

Reviewer 2: What statistical tests were used to determine significant differences in metabolic activity and ARG levels?

Author Response: Mann Whitney U test for two groups with Bonferroni correction for multiple hypothesis testing this was updated in the methods section to accurately reflect the statistical methods utilized (p. 6, lines 152-154)

Reviewer 2: What is the "reversal period"?

Author Response: Thank you for pointing this out, this was removed from the methods section as it is not in reference to the methods utilized in this study.

Reviewer 2: mitochondrial?

Author Response: This was removed from the methods section as it was inadvertently added. The text now reads "redox reduction due to respiration in the presence of a 4 graded increasing concentration of 48 different antibiotics."

Reviewer 2: Lines 175-181 are a repeat of 165-171.

Author Response: This section was inadvertently repeated and consolidated into a single section.

Reviewer 2: Line 185: Should say "the mouse genome" and "Non-mouse reads," right?
How were differentially abundant ARGs identified?

Author Response: Yes the methods were updated for non-mouse reads as pointed out. The differential abundance ARGs were further clarified on p.7 lines 178-181

“Antimicrobial resistance genes (ARGs) were predicted from assembled metagenomic contigs using Abricate v12 [24]. Abricate was run against the CARD database[23]. The parameters were set at --sensitive, --min-id 80, --min-cov 90. Results were filtered to include only hits with a minimum alignment length of 100 bp. This approach provided a comprehensive and robust assessment of ARG diversity within the metagenome. The output was parsed to generate a summary table containing information on the identified ARGs, including contig ID, gene name, database source, percentage identity, percentage coverage, and alignment length.”

Reviewer 2: Suggestions to make the figures clearer/more impactful: Consider putting the antibiotics in figures 1A and 1B in the same order so that it is easier to compare between anaerobic and aerobic. Similarly, ordering the antibiotic classes in 2A and 2B the same and the ARGs in 4A and 4B the same, as much as the overlap allows, would facilitate comparison.

Author Response: Thank you to the reviewer for the suggestions on the figures, but the current order of the antibiotics is based on which antibiotics are significantly different between WD and SD mice. By changing the order, in our opinion it is more challenging to assess the antibiotics and antibiotic classes that are statistically different between the two groups.

Reviewer 2: I'd recommend keeping the order of aerobic and anaerobic results the same in Figures 1 and 2 (ie. Have figure 1A and 2A both be anaerobic results and 1B and 2B both be aerobic results). I'd also recommend a consistent (and color-blind safe) color scheme for the 4 sample types (WD stool, WD cecum, SD stool, SD cecum) throughout all figures.

Author Response: Thank you for the suggestions to improve the figures. The figures have been altered so aerobic and anaerobic were in the same order between Figures 1 and Figures 2. The color schemes have been updated to be consistent throughout all the figures.

Reviewer 2: Line 223: It looks like 13 (not 16) are significant in Figure S2.

Author Response: Thank you for noting this, the text was updated.

Reviewer 2: Line 228: Figure 1D indicates that the higher growth from WD with neomycin is not significant.

Author Response: Thank you for noting this, the text was updated to reflect that amikacin, cefoxitin and ofloxacin were significant and neomycin was a non-significant increase to more accurately reflect the findings.

“The WD cecal microbiota demonstrated a significant increase in growth in the presence of amikacin, cefoxitin, and ofloxacin. Additionally, WD cecal microbiota had an increase in growth in the presence of neomycin that did not reach significance”

Reviewer 2: Line 241: Rather than increased beta diversity with WD it is probably more accurate to say altered beta diversity.

Author Response: The text was updated as suggested to reflect 'altered beta diversity' as opposed to reduced.

Reviewer 2: Line 248: Figure 4A shows 4 (not 6) ARGs that differ by diet in the cecum.

Author Response: The reviewer is correct there were 4 ARGs within the cecal microbiota and 6 ARGs within the stool microbiota of WD fed mice. The text was updated to accurately reflect the number of ARGs in the cecum and stool microbiota in WD mice.

Reviewer 2: Lines 249 and 526: This is relative abundance not abundance, right?

Author Response: This is correct and the text was updated to include relative abundance

Reviewer 2: Line 251-252: According to Figure 4, the ARGs that were higher in stool were not all the same as those higher in cecum.

Author Response: This is correct, the ARGs CfxA2, ErmG, and melA were shared between the stool and cecal microbiota of WD fed mice so there were some overlaps in ARGs between the groups.

Reviewer 2: Line 258: Please clarify that there are 6 ARGs increased with WD in the stool and cecum combined.

Author Response: The text was updated to include "there are 6 ARGs that are significantly increased in the WD stool and cecal microbiota without prior exposure to antibiotics."

Reviewer 2: Lines 266 and 268: I don't think gene "expression" was measured, but rather gene levels.

Author Response: The text was updated to state gene levels instead of expression.

What do the error bars in figures 1, 2, 3A and 4C represent? Should there be error bars on figs 3C&D, 4A&B? How many mice are represented in the metagenomic figures (3 and 4A&B)?

Author Response: The error bars in figure 1,2,3A, and 4C represent standard deviation, the figure legends were updated to state this. Metagenomics included 5 mice per group (5 SD, 5 WD) as stated in the figure legend.

Reviewer 2: What was the metagenomic sequencing depth? Were any samples excluded due to low sequence counts?

Author Response: The average sequencing read count was 74,754,957 reads per sample. None of the samples needed to be excluded due to low sequencing.

Reviewer 2: Were other ARGs (besides those in Figure 4) detected?

Author Response: The only ARGs detected that were statistically significant on analysis were listed in Figure 4.

Reviewer 2: Figure 4A and 4B: Are these samples after 7.5 weeks on the specific diet (so in 13.5 week old mice)?

Author Response: Yes this is correct, the mice were started on their respective diet at 6 weeks old and the experiments were completed at 13.5 weeks of age.

Reviewer 2: For Figure 4C, why were day 1 and day 7 selected if the animals were fed different diets for 7.5 weeks? I don't think it is required but it would be interesting to perform qPCR on day 0 and 7.5 week samples too. Why was tetW selected for qPCR (Figure 4C) but not represented in Figure 4A or 4B?

Author Response: Thank you for this question, Day 1 and 7 were utilized to determine how quickly ARGs could be detected after mice were started on a WD diet. Agree that it would be interesting to perform on 7.5 weeks but will not be able to be performed for this paper due to sample availability. The inclusion of tetW was included because on metagenomics it was approaching significance and added another common antibiotic resistance genes that could be further compared between SD and WD fed mice by PCR.

The metagenomic sequence data is available from the NCBI's SRA (PRJNA1148150). However, the diet and timepoint for each sample is needed for reproducibility and usefulness (either in the SRA record or as a supplemental table in the manuscript).

Author Response: Thank you for pointing this out, the sample metadata will be added as a supplemental table to the manuscript. See supplemental table 2

It would be interesting to include a discussion of how the metagenomic results fit with the metabolic data. For example, *mefA* and *mel* are both at higher relative abundances with WD which fits with greater (although not significantly greater) metabolic activity from WD with macrolide exposure because "Mel, a homolog of MsrA, is an ABC-F subfamily protein associated with macrolide resistance. It is expressed on the same operon as *mefA* and *mefE*, both MFS-type efflux proteins that confer macrolide resistance."
(from <https://card.mcmaster.ca/ontology/36910>)

Author Response: Thank you for pointing this out as this is a significant strength of the paper. The antibiotic resistance profile correlated between both metagenomic sequencing and a functional metabolic assay (Biolog). A paragraph was added to the discussion pointing on the consistencies between Biolog and metagenomic sequencing (p. 11 lines 285-294)

Reviewer 2: Line 263: I think it should say: "with complete absence of the WD-associated ARGs in the cecum"

Author Response: The text was updated to reflect these changes.

Minor:

Reviewer 2: Line 272: Should "merdei" be "merdae"?

Reviewer 2: Line 293: Should "Parabacteroidetes" be "Parabacteroides"?

Reviewer 2: Line 314: antibiotics to antibiotic

Author Response: Corrected as suggested

Re: Spectrum02766-24R1 (Dietary impact on the gut resistome: western diet independently increases the prevalence of antibiotic resistance genes within the gut microbiota)

Dear Dr. Robert Charles Keskey:

Thank you for the privilege of reviewing your work. Below you will find my comments, instructions from the Spectrum editorial office, and the reviewer comments.

Revision Guidelines

Sincerely,
Jinxin Liu
Editor
Microbiology Spectrum

Reviewer #2 (Comments for the Author):

Thanks to the authors for addressing my comments. I have a couple remaining and additional comments.

Line 140: I am not familiar with what "4 graded" means in this context. Please clarify this line.

Line 178-185: Thanks for the explanation of how ARG identification was performed. I realize that my original question wasn't clear. I was wondering how it was determined which ARGs were significantly different between groups?

Line 180: I don't think 24 is the correct reference for Abriicate. See: <https://github.com/tseemann/abriicate?tab=readme-ov-file#citation>

Thanks for sharing the average sequencing depth in the response. I think it would be helpful to include that with a range or the median with an interquartile range in the manuscript text.

I appreciate the change to a more consistent color scheme. If possible, I'd recommend extending it to Figure 3B. Shapes could distinguish cecum from stool.

Which beta diversity metric is represented in Figure 3B?

Line 291: "on metagenomic" to "by metagenomics"

Reviewer #2 (Comments for the Author): Thanks to the authors for addressing my comments. I have a couple remaining and additional comments. Line 140: I am not familiar with what "4 graded" means in this context. Please clarify this line.

Response: This is meant that there are 4 different concentrations of antibiotics on the Biolog plate that increases from low to high this was rewritten to clarify this point in the text.

Reviewer: Line 178-185: Thanks for the explanation of how ARG identification was performed. I realize that my original question wasn't clear. I was wondering how it was determined which ARGs were significantly different between groups?

Response: The difference in ARG between groups was determined utilizing a pairwise comparisons in the ARG analysis with White's non-parametric t-test. This was updated in the text p.5 lines 185-186.

Reviewer: Line 180: I don't think 24 is the correct reference for Abricate.
See: <https://github.com/tseemann/abricate?tab=readme-ov-file#citation>

Response: Thank you for catching this – we updated the citation for abricate as suggested

Reviewer: Thanks for sharing the average sequencing depth in the response. I think it would be helpful to include that with a range or the median with an interquartile range in the manuscript text.

Response: We have added the sequencing depth median (73,742,246 +/- 10,836,546)

Reviewer: I appreciate the change to a more consistent color scheme. If possible, I'd recommend extending it to Figure 3B. Shapes could distinguish cecum from stool.

Response: Unfortunately, these updates were not feasible at this time, but we appreciate the reviewers feedback.

Reviewer: Which beta diversity metric is represented in Figure 3B?

Response: Bray-Curtis dissimilarity. The text was updated to reflect this in the figure legend for Figure 3.

Reviewer: Line 291: "on metagenomic" to "by metagenomics"

Response: Corrected as suggested

Re: Spectrum02766-24R2 (Dietary impact on the gut resistome: western diet independently increases the prevalence of antibiotic resistance genes within the gut microbiota)

Dear Dr. Robert Charles Keskey:

Your manuscript has been accepted, and I am forwarding it to the ASM production staff for publication. Your paper will first be checked to make sure all elements meet the technical requirements. ASM staff will contact you if anything needs to be revised before copyediting and production can begin. Otherwise, you will be notified when your proofs are ready to be viewed.

Sincerely,
Jinxin Liu
Editor
Microbiology Spectrum